# BINARY SEARCH FOR RLVR

## ABSTRACT

Reinforcement Learning from Verifiable Rewards (RLVR) is a powerful paradigm, yet it suffers from a critical inefficiency: the profound underutilization of rare successes. The challenge is two-fold: exploring to *find* a successful trajectory, and then *learning* effectively from it. While many methods focus on the former, we address the latter. For challenging tasks where rewards are awarded only to complete, successful trajectories, such successes are rare. A single such trajectory is therefore a goldmine of information, but conventional methods treat it as just one data point, wasting a crucial learning opportunity. We introduce Binary Attribution of Sparse Signals (BASS), a method that reframes the problem from finding successes to maximizing the learning extracted from them. BASS treats a verified successful trajectory not as an answer, but as a *blueprint* to be deconstructed. It performs a binary search over the trajectory's prefixes to locate the model's *edge of competence*, i.e., the boundary where correct reasoning can falter. This process unlocks the full value of a single success by generating a rich, contrastive group of *near-miss* negatives (failures from good prefixes) and *far-reach* positives (diverse successes from shorter prefixes), providing the nuanced feedback required for robust policy optimization. Unlike methods focused on proactive exploration, BASS is a reactive, credit-focused mechanism that ensures every hard-won success is maximally leveraged to sharpen the policy. On average across three challenging math benchmarks with Qwen3-8B, BASS improves the avg@32 score by $+2.7$ percentage points (pp) over the GRPO baseline, demonstrating that meticulous learning from rare successes leads to more robust and generalizable reasoning.

## 1 INTRODUCTION

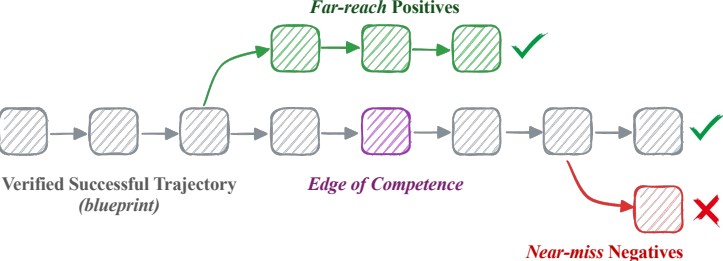

Figure 1: A verified successful trajectory acts as a blueprint to probe the model's *edge of competence*. This process generates *near-miss negatives* to pinpoint subtle failure modes, and *far-reach positives* to encourage diverse and robust solutions.

Reinforcement learning with verifiable rewards (RLVR) (Lambert et al., 2024) has emerged as a powerful paradigm for improving the reasoning abilities of Large Language Models (LLMs). Methods like Group Relative Policy Optimization (GRPO) (Shao et al., 2024) have advanced the state-of-the-art by generating multiple outcomes to construct on-the-fly baselines. However, this group-based approach suffers from a critical inefficiency: the profound underutilization of rare successes. This inefficiency is particularly acute for complex problems where a successful outcome is a rare event, making a single successful trajectory a goldmine of information. Yet conventional methods treat it

as just one single data point, failing to extract and learn the rich, sequential information within it and thus wasting a crucial learning opportunity.

While many methods focus on *finding* rare successes (Zheng et al., 2025; Hou et al., 2025), we address the subsequent, equally critical challenge of *learning* from them by treating each success not as a single data point, but as a *blueprint* for targeted exploration. We introduce Binary Attribution of Sparse Signals (BASS), a method that reframes the problem from finding successes to maximizing the learning that they provide. BASS replaces the passive acceptance of a successful outcome with an active, credit-focused mechanism. It treats a verified successful trajectory not as a final answer, but as a blueprint for exploration. By performing an efficient binary search over the trajectory's prefixes, BASS locates the model's *edge of competence*, the conceptual boundary where its reasoning is most likely to falter. This process amplifies a single data point into a rich, contrastive group of *near-miss* negatives (failures from good prefixes that pinpoint subtle failure modes) and *far-reach* positives (diverse successes from shorter prefixes that demonstrate robustness), providing a far stronger and denser signal for policy optimization.

The benefits of this principled signal amplification are clear: BASS transforms each rare success from a single data point into a rich learning opportunity, addressing the critical underutilization of positive examples. Instead of relying on entropy-driven (Cui et al., 2025; Zheng et al., 2025) heuristics to search for uncertainty, BASS reactively capitalizes on known-good states to create high-quality, contrastive training data on demand. By turning sparse successes into dense learning events, BASS provides a simple yet powerful component for existing RLVR pipelines that significantly enhances the training signal. Our work challenges the prevailing view that successes are merely endpoints, highlighting a path where intelligent, reactive credit assignment, not just brute-force exploration, drives more robust and generalizable reasoning. This results in an average improvement of $+2.7$ percentage points (pp) in the avg@32 score over the GRPO baseline on challenging math benchmarks.

## 2 RELATED WORK

Reinforcement Learning from Verifiable Rewards (RLVR) is a powerful paradigm for enhancing the reasoning abilities of language models. However, its effectiveness is often challenged by the profound sparsity of successful outcomes. In complex reasoning tasks, trajectories that achieve a positive reward are rare, making each success an extremely valuable but underutilized learning opportunity. Our work introduces a new mechanism to maximize the learning from these rare events. We position our approach relative to existing work in structured exploration, data curation, and policy dynamics.

*Structured Exploration.* A key area of research focuses on creating denser reward signals by actively exploring the structure of reasoning paths. For example, FR3E (First Return, Entropy-Eliciting Explore) uses token-level entropy to identify points of high uncertainty and launches targeted rollouts from these points to gather localized feedback (Zheng et al., 2025). Similarly, TreeRL integrates on-policy tree search into the learning loop, branching from high-entropy tokens to build a search tree and derive process-level rewards from its structure (Hou et al., 2025). Other research reinforces this idea, showing that a small number of high-entropy "forking tokens" are disproportionately important for learning (Wang et al., 2025). These methods proactively search for informative states to generate better training data, but they depend on entropy as a proxy for importance.

*Systems and Data Curation Recipes.* Another line of work adopts a data-centric perspective, addressing the sparse reward problem by carefully managing the training data distribution. POLARIS, for instance, proposes a complete post-training recipe that includes calibrating data difficulty to match model capability, using diversity-based rollouts, and applying length extrapolation at inference (An et al., 2025). Likewise, AceReason-Nemotron focuses on difficulty-aware data curation to ensure training signals remain in a learnable range (Chen et al., 2025). These system-level approaches create favorable conditions for RL to succeed, but they do not offer a specific mechanism for amplifying the signal from an individual successful trajectory once it is found.

*Policy Optimization and Entropy Management.* Research into the underlying dynamics of policy optimization reveals challenges that arise from learning on sparse data. A notable finding is the phenomenon of "policy entropy collapse", where a model's policy rapidly becomes deterministic, which stifles exploration and causes performance to stagnate (Cui et al., 2025). This work explains

that entropy dynamics are linked to the covariance between action probabilities and their advantages. This provides a theoretical basis for understanding why learning can stall and motivates methods that can provide richer, more contrastive training signals to maintain a healthy and exploratory policy. In essence, while their work illuminates the problem of how to explore to *find* successes, BASS solves the subsequent, equally critical problem of how to *learn* from them by transforming a single sparse signal into a dense, contrastive learning event.

*Positioning of BASS.* Our method, Binary Attribution of Sparse Signals (BASS), directly confronts the challenge of learning from rare successes. In RLVR, a successful trajectory is a critical learning signal, yet its value is often squandered when treated as a single data point. BASS is designed specifically to address this underutilization by treating a verified success not as an answer, but as a *blueprint* for generating a dense and informative training batch. Unlike entropy-driven methods such as FR3E and TreeRL, BASS is entropy-free; it reactively capitalizes on a known success rather than proactively searching for uncertainty. It provides a lightweight, credit-focused alternative to full tree-expansion, with the goal of efficiently localizing pivotal steps within a confirmed successful path. This process naturally generates a rich group of *near-miss* negatives and *far-reach* positives, transforming a single sparse reward into a dense, high-contrast learning signal. This makes BASS a simple yet powerful component that complements existing RLVR pipelines by ensuring that every rare success is maximally leveraged for policy improvement.

## 3 METHOD

We introduce Binary Attribution of Sparse Signals (BASS), a method that confronts the fundamental credit assignment problem for RLVR. The name is an acronym for "*B*inary search over a verified reasoning trajectory to *A*ttribute success or failure to specific reasoning steps, a connection typically obscured by a single *S*parse *S*ignal." This search acts as an adaptive sampling strategy to efficiently discover pivotal regions in the reasoning chain. The ultimate objective is not to precisely pinpoint a single point of failure, but to generate a rich *group* of trajectories. This group, composed of *near-miss* negatives and *far-reach* positives, transforms the original sparse reward into a dense and highly informative contrastive learning signal for policy optimization.

---

**Algorithm 1** Binary Attribution of Sparse Signals

**Require:** Prompt $x$, verified trajectory $y^\star = (y_1, \ldots, y_N)$ of length $N$ where we denote a prefix as $y_{1:m}^\star = (y_1, \ldots, y_m)$, with $r(x, y^\star) = 1$, iterations $K$.
**Ensure:** A group $\mathcal{G}$ of sampled trajectories.
1: $\ell \leftarrow 0, h \leftarrow N$
2: $\mathcal{G} \leftarrow \varnothing$
3: **for** $k = 1, \ldots, K$ **do**
4:      $m \leftarrow \ell + \lfloor (h - \ell)/2 \rfloor$
5:      Sample continuation $y_{\text{cont}} \sim \pi_\theta(\cdot \mid x, y_{1:m}^\star)$
6:      Form full trajectory $\tilde{y} \leftarrow y_{1:m}^\star \circ y_{\text{cont}}$
7:      $\tilde{r} \leftarrow r(x, \tilde{y})$
8:      $\mathcal{G} \leftarrow \mathcal{G} \cup \{(\tilde{y}, \tilde{r})\}$
9:      **if** $\tilde{r} = 1$ **then**
10:         $h \leftarrow m$
11:      **else**
12:         $\ell \leftarrow m + 1$
13:      **end if**
14: **end for**
15: **return** $\mathcal{G}$

---

### 3.1 BINARY ATTRIBUTION OF SPARSE SIGNALS (BASS)

The BASS procedure (Algorithm 1) takes three inputs: a prompt $x$, a corresponding verified trajectory $y^\star = y_{1:N}^\star$ where $r(x, y^\star) = 1$, and the number of search iterations, $K$. The algorithm performs a *binary search* over the prefixes of $y^\star$ to guide the generation of informative trajectories. Crucially, the single sample drawn at each midpoint is not intended as a high-fidelity evaluation of the prefix; rather, it serves as a *stochastic probe*. While a single rollout has high variance, its outcome provides a sufficiently strong signal to guide the search toward the most fruitful regions for generating contrastive data. A success suggests exploring shorter prefixes, while a failure points toward longer ones. This adaptive process populates a group $\mathcal{G}$ with both successful and failed continuations from meaningful prefixes, creating the rich contrastive data needed for effective policy learning.

### 3.2 THEORETICAL ANALYSIS OF BASS

We formally analyze the efficiency of BASS by breaking down its advantages into two core components. We validate these against two simpler baselines over $K$ sampling iterations, illustrated in Figure 2:

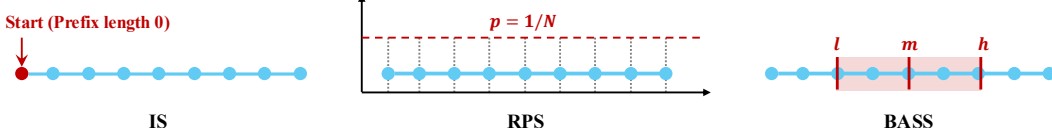

Figure 2: A visual comparison of the sampling strategies. (Left) Independent Sampling (IS) always generates continuations from the start (prefix length 0). (Center) Random Prefix Sampling (RPS) selects a prefix start point uniformly at random from $\{0, \ldots, N-1\}$ (each with probability $1/N$), which is often inefficient. (Right) BASS performs an adaptive binary search, maintaining a window $[l, h]$ and sampling from the midpoint $m$ to efficiently locate the critical reasoning boundary.

1. *Independent Sampling (IS):* All $K$ samples are generated from the initial prompt $x$ (i.e., prefix length 0).

2. *Random Prefix Sampling (RPS):* In each iteration, a prefix length is chosen uniformly at random from $\{0, \ldots, N-1\}$ to generate a continuation.

Our analysis first establishes the probabilistic benefit of restarting from a verified prefix, which justifies the generation of *far-reach* positives. Second, we prove the exponential efficiency of BASS's search strategy in locating the critical reasoning boundary, which underpins its ability to generate high-value *near-miss* negatives.

### 3.2.1 PROBABILISTIC ADVANTAGE OF PREFIX-BASED RESAMPLING

*The Gated Reasoning Model.* We first analyze why restarting from any part of a successful trajectory is inherently more promising than starting from scratch. We model a successful trajectory as one that passes $M$ independent critical gates $G_1, \ldots, G_M$, each with a pass probability of $g \in (0, 1)$. Let $M_{\text{pref}}(\alpha)$ be the number of gates located within the first $\alpha$ fraction of a verified trajectory's tokens. Independent sampling succeeds with probability $P_{\text{ind}} = g^M$. In contrast, restarting from a prefix of fraction $\alpha$ bypasses the gates already passed, succeeding with probability $P_{\text{restart}} = g^{M-M_{\text{pref}}(\alpha)}$. This yields a probabilistic improvement factor of $\rho(\alpha) = g^{-M_{\text{pref}}(\alpha)}$. While this is a simplified abstraction of the complex, sequential nature of LLM reasoning, it provides a tractable framework for analyzing the probabilistic benefits of resampling from a known-good prefix.

**Lemma 3.1 (Distribution-free lower bound).** *If $\Pr[M_{\text{pref}}(\alpha) \geqslant 1] \geqslant p$, then $\mathbb{E}[\rho(\alpha)] \geqslant 1 + (g^{-1} - 1)p$.*

*Proof.* We can find a lower bound for the expected improvement factor, $\mathbb{E}[\rho(\alpha)]$, by conditioning on whether any critical gates have been passed. Let $A$ be the event that $M_{\text{pref}}(\alpha) \geqslant 1$, which occurs with probability $\Pr[A] \geqslant p$. By the law of total expectation, $\mathbb{E}[\rho(\alpha)] = \mathbb{E}[\rho(\alpha)|A]\Pr[A] + \mathbb{E}[\rho(\alpha)|A^c]\Pr[A^c]$. If event $A$ occurs, $M_{\text{pref}}(\alpha) \geqslant 1$, so the improvement factor $\rho(\alpha) = g^{-M_{\text{pref}}(\alpha)} \geqslant g^{-1}$ since $g < 1$. Thus, $\mathbb{E}[\rho(\alpha)|A] \geqslant g^{-1}$. If $A$ does not occur (event $A^c$), then $M_{\text{pref}}(\alpha) = 0$, and $\rho(\alpha) = g^0 = 1$. Substituting these into the formula yields $\mathbb{E}[\rho(\alpha)] \geqslant g^{-1}\Pr[A] + 1 \cdot (1 - \Pr[A]) = 1 + (g^{-1} - 1)\Pr[A]$. Because $g^{-1} > 1$, this expression is increasing with $\Pr[A]$. Using the given condition $\Pr[A] \geqslant p$, we arrive at the final lower bound, $1 + (g^{-1} - 1)p$. $\qquad\square$

To quantify the advantage more precisely, we introduce the *uniform placement model*. This model, while a simplification of the true, often complex distribution of reasoning steps, provides a tractable framework for analysis. We assume that the critical gates are distributed uniformly throughout the trajectory's generation process. This implies that the number of gates in a prefix of fraction $\alpha$ can be modeled by a binomial distribution, $M_{\text{pref}}(\alpha) \sim \text{Binom}(M, \alpha)$.

**Lemma 3.2 (Uniform-placement model).** *Under the uniform placement model, the expected improvement factor is $\mathbb{E}[\rho(\alpha)] = \left((1-\alpha) + \alpha g^{-1}\right)^M$.*

*Proof.* Under this model, the number of bypassed gates, $X = M_{\text{pref}}(\alpha)$, follows a binomial distribution, $X \sim \text{Binom}(M, \alpha)$. Our goal is to compute the expectation of the improvement factor, $\mathbb{E}[\rho(\alpha)] = \mathbb{E}[g^{-X}]$. This expression can be rewritten as $\mathbb{E}[(g^{-1})^X]$. This is the definition

of the Probability Generating Function (PGF) of the random variable $X$, evaluated at the point $z = g^{-1}$. The PGF for a binomial random variable $X \sim \mathrm{Binom}(n, p)$ is given by the formula $G_X(z) = \mathbb{E}[z^X] = (1 - p + pz)^n$. In our specific case, the parameters are $n = M$ and $p = \alpha$. By substituting these and $z = g^{-1}$ into the PGF formula, we directly obtain the expected improvement factor: $\mathbb{E}[\rho(\alpha)] = ((1 - \alpha) + \alpha g^{-1})^M$. This result quantifies the precise probabilistic advantage when gates are distributed uniformly. $\qquad\square$

This analysis confirms that leveraging a verified prefix offers a substantial probabilistic advantage over independent sampling, a finding we validate in our first experiment (§4.1). However, this model does not specify *which* prefixes are most informative. Randomly choosing them is inefficient, which motivates our next point.

### 3.2.2 EXPONENTIAL EFFICIENCY OF ADAPTIVE SEARCH

*The Pivotal Step Model.* To analyze the efficiency of the *search strategy* itself, we now address how BASS intelligently selects prefixes. To isolate and analyze the efficiency of the search component, we adopt a deterministic *pivotal step model*. While unrealistic, this model allows for a clear comparison of search strategies by assuming success hinges on a single "pivotal" step at index $c^\star \in \{1, \ldots, N\}$. A continuation from prefix $y_{1:m}^\star$ succeeds if and only if $m \geqslant c^\star$. The most informative samples for learning are those generated from prefixes near this critical boundary. We define a *high-information zone* $I_\delta = \{m : |m - (c^\star - 1/2)| \leqslant \delta\}$, centered around the pivotal step (the $1/2$ term centers the window between integers), and compare how efficiently BASS and RPS can concentrate samples within this zone.

> **Proposition 3.3** (**BASS Exponentially Outperforms Random Search**). *Let $N_I^{rand}$ and $N_I^{BASS}$ be the number of samples from $I_\delta$ for RPS and BASS.*
>
> *1. For Random Prefix Sampling, the expected number of informative samples is:*
> $$\mathbb{E}[N_I^{rand}] = K \cdot \frac{|I_\delta \cap \{0, \ldots, N-1\}|}{N} \approx K \cdot \frac{2\delta}{N}$$
>
> *2. For BASS, the number of informative samples has a lower bound:*
> $$\mathbb{E}[N_I^{BASS}] \geqslant \max\left(0, K - \lceil \log_2(N/\delta) \rceil\right)$$

*Proof.* 1. Each sample $m_k$ is an independent uniform draw. The probability of any single sample falling into $I_\delta$ is $p \approx 2\delta/N$. By linearity of expectation over $K$ trials, $\mathbb{E}[N_I^{rand}] = K \cdot p$.

2. BASS maintains a search window $[\ell, h]$ guaranteed to contain $c^\star$, with size $W_k = h - \ell \approx N/2^k$ at iteration $k$. The sample point $m_k$ is the midpoint of this window. A sample $m_k$ is guaranteed to fall within $I_\delta$ if the entire search window's half-width is smaller than the zone's width, i.e., $W_k/2 \leqslant \delta$, as this ensures $|m_k - c^\star| \leqslant W_k/2 \leqslant \delta$. This condition is met once the window size $W_k$ is sufficiently small, which occurs after approximately $k_0 = \lceil \log_2(N/\delta) \rceil$ iterations. For all subsequent $k > k_0$, every sample is guaranteed to be informative. The bound follows. $\qquad\square$

This analysis proves that BASS's adaptive search is exponentially more efficient than random search at finding the critical boundary. We validate this theoretical finding with a controlled simulation in our second experiment (§4.2).

*Synthesis and Implications.* Our two-part theoretical analysis provides a complete justification for BASS. The *gated reasoning model* shows that restarting from verified prefixes dramatically increases the probability of success. The *pivotal step model* proves that BASS's binary search is an exponentially efficient method for finding the *most valuable* prefixes to restart from. Together, these properties allow BASS to transform a single sparse reward into a dense and powerful contrastive learning signal.

### 3.3 COMPUTATIONAL COST AS AN INVESTMENT IN SIGNAL QUALITY

The computational overhead of BASS, i.e., the $K$ additional rollouts for its binary search, should be viewed not as a cost but as a strategic investment in signal quality. Consider the core scenario:

for a difficult prompt, initial sampling yields a sparse group of size $N_g$ (e.g., $N_g = 8$) with one success and $N_g - 1$ failures. While this group, $\mathcal{G}_{\text{init}}$, is technically trainable for GRPO, it is highly inefficient. The single success is underutilized, and the uninformative failures provide a noisy, low-quality learning signal.

We can formalize this inefficiency through the lens of *policy gradient optimization*. The gradient estimator $\hat{g}$ for a group $\mathcal{G}$ relies on the advantages $A(\tau_i)$ and score functions $\nabla_\theta \log P(\tau_i; \theta)$ of its trajectories. For the initial group $\mathcal{G}_{\text{init}} = \{\tau^*\} \cup \{\tau_{\text{fail},i}\}_{i=1}^{N_g - 1}$, containing one success $\tau^*$ and $N_g - 1$ failures, the gradient estimate is:

$$\hat{g}_{\text{init}} = \frac{1}{N_g} \left( A(\tau^*)\nabla_\theta \log P(\tau^*) + \sum_{i=1}^{N_g - 1} A(\tau_{\text{fail},i})\nabla_\theta \log P(\tau_{\text{fail},i}) \right). \tag{1}$$

The critical weakness lies in the high variance of the score functions $\nabla_\theta \log P(\tau_{\text{fail},i})$ for the uninformative failures. These trajectories can be wildly heterogeneous, causing their score vectors to point in diverse directions in the parameter space. This results in a $\hat{g}_{\text{init}}$ with a very poor signal-to-noise ratio.

BASS offers a fundamentally different approach. It discards the noisy failures and makes a fixed, predictable investment of $K$ rollouts to generate a new group $\mathcal{G}_{\text{bass}}$ from the single success. This group is composed of structurally similar *near-miss* negatives and *far-reach* positives.

> **Claim 3.4** (**Lower Gradient Variance for BASS**). *The gradient estimator $\hat{g}_{bass}$, derived from the BASS-generated group, has a significantly lower variance than $\hat{g}_{init}$.*

*Justification.* This claim is based on the geometric properties of the score function space. By construction, trajectories in $\mathcal{G}_{\text{bass}}$ (e.g., a near-miss negative $\tau_{\text{nm}}$) share long prefixes with the original success $\tau^*$. Consequently, their score function vectors, $\nabla_\theta \log P(\tau)$, are highly correlated as they are identical for the shared prefix. This structural homogeneity ensures that the set of score vectors within $\hat{g}_{\text{bass}}$ has low intrinsic variance. In contrast, the uncorrelated, heterogeneous failures in $\mathcal{G}_{\text{init}}$ produce a high-variance set of score vectors. A lower-variance gradient estimate leads to more stable and precise policy updates, focusing learning directly on the critical reasoning steps at the *edge of competence*. $\square$

Therefore, the computational overhead of BASS is a well-justified trade-off. It replaces a wasteful learning process on a noisy signal with a targeted, fixed-cost procedure that generates a high-quality, low-variance gradient estimator. This fundamentally more efficient allocation of resources unlocks the full learning potential of every rare success.

### 3.4 PRACTICAL INTEGRATION WITH GRPO

The group of trajectories $\mathcal{G}$ generated by BASS is naturally suited for policy optimization methods like GRPO (Shao et al., 2024; DeepSeek, 2025), which derive their learning signal from reward variance within a group. A common scenario in GRPO training is the emergence of "zero-advantage" groups, where all trajectories for a given prompt share the same reward (i.e., all successes or all failures). Such groups provide no contrastive signal for policy optimization, as the advantage estimate for every trajectory within them is zero. By selecting a successful trajectory from a group with a low success rate (e.g., containing only one or two successes in a group of eight), BASS can transform it into a new, high-contrast group $\mathcal{G}$ containing both *near-miss* negatives and *far-reach* positives. This targeted application not only enhances the training signal but also allows the computational investment of BASS to be amortized, as detailed in our integration strategy.

Our integration strategy consists of two main components:

1. *Memory Buffer for Rare Trajectories:* We maintain a memory buffer $\mathcal{M}$ that stores trajectories from prompts where the model is on the cusp of competence (i.e., prompts that yield rare successes). These samples, containing nascent versions of a verified trajectory, are accumulated in $\mathcal{M}$ across training steps.

2. *Periodic Batch Augmentation:* At a fixed interval of training steps, we improve the quality of the current training batch. We identify zero-advantage groups (where all rollouts either

succeed or fail) and replace them with high-contrast data generated by BASS. This is done by sampling a successful trajectory $y^\star$ from our buffer $\mathcal{M}$, running BASS to generate a rich group $\mathcal{G}$, and substituting this group for a low-signal sample in the batch.

The resulting optimized batch, containing a mix of standard rollouts and BASS-generated contrastive data, is then used for the policy update. We compute advantage estimates $\tilde{A}_t$ for all trajectories and update the policy parameters using the standard PPO-Clip policy loss (Schulman et al., 2017):

$$L^{\mathrm{CLIP}}(\theta) = \mathbb{E}_{s,t}\left[\min\left(\frac{\pi_\theta(a_t \mid s_t)}{\pi_{\theta_{\mathrm{old}}}(a_t \mid s_t)}\,\tilde{A}_t,\ \mathrm{clip}\left(\frac{\pi_\theta(a_t \mid s_t)}{\pi_{\theta_{\mathrm{old}}}(a_t \mid s_t)},\ 1-\varepsilon,\ 1+\varepsilon\right)\tilde{A}_t\right)\right]. \quad (2)$$

## 4 EXPERIMENTS

### 4.1 VALIDATING PROBABILISTIC ADVANTAGE: RELIABILITY IN GENERATING SUCCESSES

To validate the premise from our *Gated Reasoning Model*, that leveraging prefixes increases the probability of success, we test BASS's reliability in identifying stable, high-success regions in the model's generation space.

The experiment, detailed in Figure 3, compares BASS against a baseline of independent sampling across a suite of models: Qwen2.5-7B (Qwen, 2024), Qwen3-8B/30B-A3B-Thinking-2507 (Qwen, 2025), and DeepSeek-R1-Distill-Qwen-1.5B (DeepSeek, 2025). For each model, we curated 50 "hard" prompts from the Polaris dataset (An et al., 2025) where the model initially had a low $(1/8)$ success rate. After identifying one positive response, we sampled 7 more times using both methods. As shown in the figure, the baseline method (top bars) proves unreliable, with most resampling attempts yielding two or fewer successes (red bars). In stark contrast, BASS (bottom bars) consistently shifts the distribution toward higher success rates. Across all models, it produces a significantly greater number of trials with 3-7 successes, confirming its effectiveness in exploiting a known success to find and stabilize a high-success generation region.

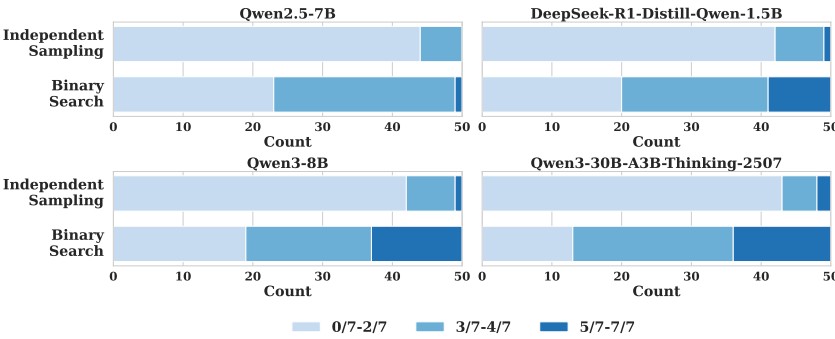

Figure 3: Our binary search heuristic (bottom row) is significantly more reliable than a baseline (top row). After finding one success, we resampled 7 times. Our method consistently yields 3–7 successful outcomes, while the baseline rarely achieves more than two, showing our method effectively locates high-success generation regions.

### 4.2 VALIDATING SEARCH EFFICIENCY: LOCATING THE EDGE OF COMPETENCE

To empirically validate the exponential efficiency gains of BASS over Random Prefix Sampling (RPS), as predicted by Proposition 3.3 and the *Pivotal Step Model*, we conduct a controlled simulation. We model a trajectory of length $N = 256$ with a single ground-truth "pivotal step" at index $c^*$. A rollout is considered successful if and only if it is generated from a prefix of length $m \geqslant c^*$. The most informative samples for learning are those generated from prefixes within a *high-information zone* surrounding $c^*$. We compare BASS against RPS over $K = 8$ sampling iterations in two scenarios: an early pivotal step $(c^* = 50)$ and a late pivotal step $(c^* = 210)$.

The results, shown in Figure 4, clearly demonstrate the superior efficiency of BASS's adaptive search strategy. In both scenarios, the samples generated by RPS (blue circles) are uniformly scattered

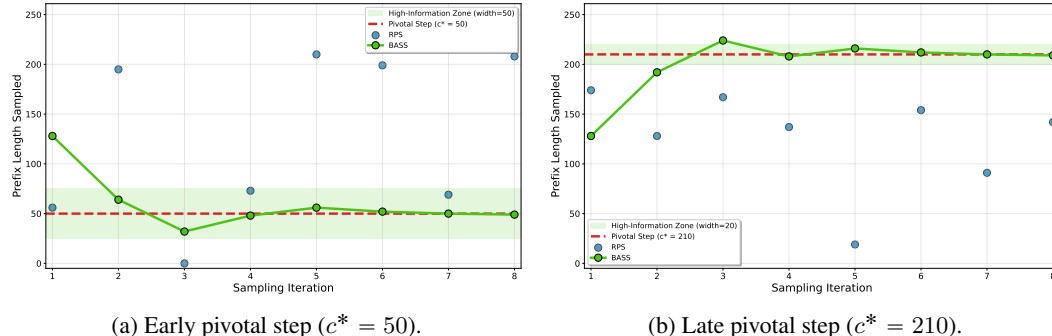

(a) Early pivotal step ($c^* = 50$).  (b) Late pivotal step ($c^* = 210$).

Figure 4: Comparison of BASS and RPS search strategies. In both an early (a) and late (b) pivotal step scenario, BASS (green line) rapidly converges to the pivotal step (red dashed line) and concentrates samples in the high-information zone (shaded area), while RPS samples (blue dots) are scattered inefficiently.

across the entire prefix space. Consequently, only a small fraction of its probes land within the high-information zone (shaded green area), and this occurs purely by chance. Most of its computational budget is wasted on uninformative prefixes that are either trivially successful or hopelessly unsuccessful.

In stark contrast, BASS (green line with markers) exhibits a systematic and rapid convergence. Regardless of whether the pivotal step is early (Figure 4a) or late (Figure 4b) in the trajectory, BASS's binary search strategy exponentially narrows the search window, focusing its samples directly on the critical boundary. Within approximately 3-4 iterations, BASS consistently samples from within the high-information zone for the remainder of the procedure.

This simulation confirms that BASS does not merely find successful continuations; it efficiently pinpoints the model's *edge of competence*. This allows it to generate a dense set of high-value contrastive samples, *near-miss* negatives and *far-reach* positives, from a single successful trajectory, directly addressing the core challenge of learning from sparse rewards.

### 4.3 ONLINE TRAINING EXPERIMENTS

The BASS algorithm is broadly applicable in LLM reasoning tasks (DeepSeek, 2025) and Agentic training. We evaluate Tool-Integrated Reasoning (TIR) (Feng et al., 2025; Lin & Xu, 2025) scenarios, where the LLMs can utilize external Python interpreter to help solve hard problems. We conduct experiments using a moderately sized LLM, Qwen3-8B (Qwen, 2025). For training data, we use the English

Table 1: Comparison of GRPO, RPS, and BASS on three benchmarks using the avg@32 metric. Bold indicates the best-performing method.

| Method | AIME 24 | AIME 25 | BeyondAIME | Average |
|--------|---------|---------|------------|---------|
| GRPO   | 74.6    | 64.1    | 38.1       | 58.9    |
| RPS    | 75.1    | 65.3    | 37.7       | 59.4    |
| BASS   | **77.8** | **67.0** | **39.9**  | **61.6** |

subset from the DAPO dataset (Yu et al., 2025). Only outcome reward is applied for RLVR, without the format rewards. We evaluate performance on the challenging math competition benchmarks, i.e., AIME 24, AIME 25, and BeyondAIME (Seed, 2025). See Appendix B for training and evaluation details.

Our experiments confirm the superiority of BASS's adaptive strategy, which significantly outperforms both the GRPO baseline and non-adaptive Random Prefix Sampling (RPS). As shown in Table 1, BASS improves the average avg@32 score to 61.6, a substantial gain of +2.7 percentage points (pp) over GRPO and +2.2 pp over RPS. This strong aggregate performance stems from BASS's consistent dominance across all individual benchmarks. It secures top scores on AIME 24 (+2.7 pp over RPS) and AIME 25 (+1.7 pp over RPS), and crucially, its advantage holds on the BeyondAIME benchmark, a dataset specifically designed to be difficult and contamination-resistant. On this true test of reasoning, BASS delivers a robust gain while the untargeted RPS approach falters and degrades performance below the baseline. This demonstrates that BASS's adaptive search for the *edge of competence* reliably transforms sparse successes into robust and generalizable solutions.

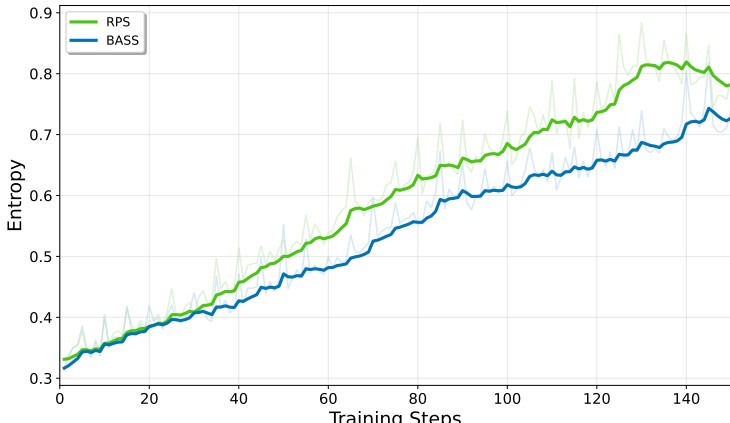

Figure 5: Policy Entropy During Training. BASS's lower entropy reflects a more focused and efficient learning process.

### 4.4 DECODING POLICY ENTROPY: FOCUSED VS. UNFOCUSED EXPLORATION

To further investigate the learning dynamics of BASS, we tracked policy entropy during training against the Random Prefix Sampling (RPS) baseline (Figure 5). The results provide strong evidence for the efficiency of BASS's targeted strategy.

*Focused Learning Signal:* BASS maintains a significantly lower policy entropy than RPS throughout training. This is not a sign of poor exploration but a direct consequence of its high-quality data generation. By focusing on the *edge of competence* to produce dense batches of informative examples, BASS provides the model with a clear, low-variance learning signal, reducing policy uncertainty and allowing it to learn correct reasoning paths with confidence.

*Unfocused Exploration:* In contrast, the higher entropy of RPS reflects its inefficiency. By sampling prefixes randomly, RPS presents the model with a high-variance mixture of uninformative states (some trivially easy, others hopelessly difficult). This lack of focus creates a noisy signal, forcing the policy to remain in a state of high uncertainty. Its higher entropy is therefore a symptom of *unfocused exploration*, not effective exploration.

In summary, the entropy analysis shows that BASS's intelligent, credit-focused mechanism achieves a more sample-efficient learning process than a brute-force, high-entropy search. The lower entropy is a hallmark of a policy learning with precision from a dense, high-quality signal.

## 5 CONCLUSIONS

We identified a critical inefficiency in existing Reinforcement Learning from Verifiable Rewards (RLVR) methods: the profound underutilization of rare successes. For complex tasks that grant rewards only at the trajectory level, successful outcomes are infrequent, making a single successful trajectory a goldmine of information. Yet conventional methods often treat it as one data point, wasting a crucial learning opportunity. To address this, we proposed Binary Attribution of Sparse Signals (BASS), a principled framework that shifts the focus from *finding* successes to *maximizing the learning extracted from them*. BASS leverages a single verified success as a blueprint, performing a binary search to systematically probe the model's reasoning boundary and generate a dense, contrastive batch of *near-miss* negatives and *far-reach* positives.

Our work demonstrates that a reactive, credit-focused approach transforms learning from rare successes into a tractable task. By amplifying each success into a high-quality, low-variance learning signal, BASS enables more stable policy optimization, recasting successes as invaluable opportunities for targeted learning. Importantly, the simplicity of the BASS algorithm makes it highly reproducible and easy to integrate into existing RLVR pipelines. Future work could explore more precise search strategies to pinpoint the edge of competence, balancing complexity against the fundamental efficiency that makes BASS a practical tool.

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

## A  LLMS USAGE

LLMs were employed solely to assist with minor editorial improvements to the text. Their use was limited to polishing language, and they did not contribute to the conceptual or scientific content of the paper.

## B  TRAINING AND EVALUATION DETAILS

Our implementation for tool-integrated reasoning leverages the ReTool (Feng et al., 2025) frameworks as its foundation. The policy is refined using the Clip-Higher mechanism (Yu et al., 2025) to carefully balance exploration and exploitation, with parameters $\varepsilon_{\text{low}}$ and $\varepsilon_{\text{high}}$ set to 0.2 and 0.28, respectively. We apply a conservative learning rate of $1 \times 10^{-6}$ for these policy updates. Trajectories for training are generated with diverse sampling settings (temperature=1.0, top-$p = 1.0$, top-$k = -1$) and are capped at a maximum length of 16,384 tokens. In terms of data throughput, our process handles a batch of 96 prompts at a time, generating 8 responses for each, which are then organized into training mini-batches of 96. Each reasoning episode permits up to 8 interactions with the external Python interpreter.

We evaluate on challenging math competition benchmarks including AIME 24, AIME 25, and BeyondAIME (Seed, 2025). Decoding is performed with temperature 0.6, top-$p$ 0.95, and top-$k$ 20, following recommended defaults. Responses are capped at 32,768 tokens. A binary reward $r_{i,j}$ is assigned to each response: $r_{i,j} = 1$ if the final answer is correct and $r_{i,j} = 0$ otherwise. The same criterion is applied in both training (for policy optimization) and evaluation. Given $M$ test problems, and $k$ sampled responses per problem with rewards $\{r_{i,1}, \ldots, r_{i,k}\}$, we report the avg@$k$ metric, which is the mean correctness across all responses: avg@$k = \frac{1}{M} \sum_{i=1}^{M} \frac{1}{k} \sum_{j=1}^{k} r_{i,j}$.

## C  FORMAL PROOFS FOR THEORETICAL ANALYSIS

This appendix provides detailed, step-by-step proofs for the lemmas and proposition presented in the theoretical analysis of BASS.

### C.1  PROOF OF LEMMA 3.1

**(Lemma 3.1 Distribution-free lower bound).** If $\Pr[M_{\text{pref}}(\alpha) \geqslant 1] \geqslant p$, then $\mathbb{E}[\rho(\alpha)] \geqslant 1 + (g^{-1} - 1)p$.

*Proof.* The proof proceeds by applying the law of total expectation to the improvement factor $\rho(\alpha)$ and then using the given probabilistic bound.

**1. Definitions and Setup**

- The improvement factor is defined as $\rho(\alpha) = g^{-M_{\text{pref}}(\alpha)}$, where $g \in (0, 1)$ is the gate pass probability and $M_{\text{pref}}(\alpha)$ is the number of gates passed in a prefix of fractional length $\alpha$.

- Let $A$ be the event that at least one gate has been passed, i.e., $A = \{M_{\text{pref}}(\alpha) \geqslant 1\}$.

- Let $A^c$ be the complement event, where no gates have been passed, i.e., $A^c = \{M_{\text{pref}}(\alpha) = 0\}$.

- We are given the premise that $\Pr[A] \geqslant p$.

**2. Law of Total Expectation** The expected improvement factor can be decomposed by conditioning on event $A$:

$$\mathbb{E}[\rho(\alpha)] = \mathbb{E}[\rho(\alpha)|A] \Pr[A] + \mathbb{E}[\rho(\alpha)|A^c] \Pr[A^c]. \tag{3}$$

**3. Bounding the Conditional Expectations**

- **Case 1 (Event $A$):** If event $A$ occurs, we know $M_{\text{pref}}(\alpha) \geqslant 1$. Since $g \in (0, 1)$, it follows that $g^{-1} > 1$. The function $f(x) = (g^{-1})^x$ is strictly increasing for $x \geqslant 0$. Therefore:

$$M_{\text{pref}}(\alpha) \geqslant 1 \implies g^{-M_{\text{pref}}(\alpha)} \geqslant g^{-1}.$$

This implies that the value of $\rho(\alpha)$ is at least $g^{-1}$ given event $A$. The expectation conditioned on $A$ is therefore also bounded:

$$\mathbb{E}[\rho(\alpha)|A] \geqslant g^{-1}.$$

- **Case 2 (Event $A^c$):** If event $A^c$ occurs, we know $M_{\text{pref}}(\alpha) = 0$. The improvement factor is deterministic in this case:
$$\rho(\alpha) = g^{-0} = 1.$$
  The conditional expectation is therefore exactly 1:
$$\mathbb{E}[\rho(\alpha)|A^c] = 1.$$

**4. Substituting Bounds and Simplifying** Substitute these lower bounds back into Equation 3:
$$\mathbb{E}[\rho(\alpha)] \geqslant (g^{-1})\Pr[A] + (1)\Pr[A^c]$$
$$= g^{-1}\Pr[A] + (1 - \Pr[A])$$
$$= 1 + (g^{-1} - 1)\Pr[A].$$

**5. Applying the Premise** We are given that $\Pr[A] \geqslant p$. Since $g^{-1} > 1$, the term $(g^{-1} - 1)$ is positive. Therefore, the expression $1 + (g^{-1} - 1)\Pr[A]$ is an increasing function of $\Pr[A]$. We can thus substitute the lower bound for $\Pr[A]$ to obtain a final lower bound on the expectation:
$$\mathbb{E}[\rho(\alpha)] \geqslant 1 + (g^{-1} - 1)p.$$

This completes the proof. $\qquad\square$

### C.2 PROOF OF LEMMA 3.2

**Lemma 3.2 (Uniform-placement model)** Under the uniform placement model, the expected improvement factor is $\mathbb{E}[\rho(\alpha)] = \left((1-\alpha) + \alpha g^{-1}\right)^M$.

*Proof.* The proof relies on the properties of the Probability Generating Function (PGF) for the binomial distribution.

**1. Model and Definitions**

- Let $X$ be the random variable for the number of gates in the prefix, $X = M_{\text{pref}}(\alpha)$.

- Under the uniform placement model, each of the $M$ gates has an independent probability $\alpha$ of being in the prefix. Thus, $X$ follows a binomial distribution: $X \sim \text{Binom}(M, \alpha)$.

- The improvement factor is a function of this random variable: $\rho(\alpha) = g^{-X}$.

**2. Calculating the Expectation using the PGF** Our goal is to compute $\mathbb{E}[\rho(\alpha)] = \mathbb{E}[g^{-X}]$. We can rewrite this expression as:
$$\mathbb{E}[g^{-X}] = \mathbb{E}[(g^{-1})^X].$$
This form matches the definition of the Probability Generating Function (PGF) of a discrete random variable $X$, which is defined as $G_X(z) = \mathbb{E}[z^X]$. Our target expectation is simply the PGF of $X$ evaluated at the point $z = g^{-1}$.

**3. Applying the Binomial PGF** The PGF for a binomial random variable $X \sim \text{Binom}(n, p)$ is given by the well-known formula:
$$G_X(z) = (1 - p + pz)^n.$$
For our specific case, the parameters are $n = M$ and $p = \alpha$. Substituting these into the formula and evaluating at $z = g^{-1}$, we get:
$$\mathbb{E}[g^{-X}] = G_X(g^{-1})$$
$$= \left(1 - \alpha + \alpha(g^{-1})\right)^M$$
$$= \left((1-\alpha) + \alpha g^{-1}\right)^M.$$

This calculation directly yields the desired expression for the expected improvement factor. This completes the proof. $\qquad\square$

## C.3 PROOF OF PROPOSITION 3.3

**Proposition 3.3 (BASS Exponentially Outperforms Random Search)** Let $N_I^{\text{rand}}$ and $N_I^{\text{BASS}}$ be the number of samples from the high-information zone $I_\delta = \{m : |m - (c^\star - 1/2)| \leqslant \delta\}$ for RPS and BASS, respectively, over $K$ iterations.

1. For Random Prefix Sampling, the expected number of informative samples is:

$$\mathbb{E}[N_I^{\text{rand}}] = K \cdot \frac{|I_\delta \cap \{0, \ldots, N-1\}|}{N} \approx K \cdot \frac{2\delta}{N}$$

2. For BASS, the number of informative samples has a lower bound:

$$\mathbb{E}[N_I^{\text{BASS}}] \geqslant \max\left(0, K - \lceil \log_2(N/\delta) \rceil\right)$$

*Proof.* We prove each part separately.

**Part 1: Random Prefix Sampling (RPS)**

- **Setup:** In each of $K$ independent iterations, a prefix length $m_k$ is drawn uniformly at random from the set $\mathcal{S} = \{0, 1, \ldots, N-1\}$.

- Let $X_k$ be an indicator random variable for the $k$-th iteration, where $X_k = 1$ if $m_k \in I_\delta$ and $X_k = 0$ otherwise. The total number of informative samples is $N_I^{\text{rand}} = \sum_{k=1}^{K} X_k$.

- **Linearity of Expectation:**

$$\mathbb{E}[N_I^{\text{rand}}] = \mathbb{E}\left[\sum_{k=1}^{K} X_k\right] = \sum_{k=1}^{K} \mathbb{E}[X_k].$$

- **Expectation of an Indicator:** The expectation of an indicator variable is the probability of the event it indicates: $\mathbb{E}[X_k] = \Pr[X_k = 1] = \Pr[m_k \in I_\delta]$.

- **Probability Calculation:** Since $m_k$ is drawn uniformly from $\mathcal{S}$, the probability is the ratio of the number of favorable outcomes to the total number of outcomes:

$$\Pr[m_k \in I_\delta] = \frac{|\mathcal{S} \cap I_\delta|}{|\mathcal{S}|} = \frac{|I_\delta \cap \{0, \ldots, N-1\}|}{N}.$$

- **Final Result:** Since each trial is identical, $\mathbb{E}[X_k]$ is the same for all $k$.

$$\mathbb{E}[N_I^{\text{rand}}] = \sum_{k=1}^{K} \frac{|I_\delta \cap \{0, \ldots, N-1\}|}{N} = K \cdot \frac{|I_\delta \cap \{0, \ldots, N-1\}|}{N}.$$

  The interval $I_\delta$ has a width of $2\delta$, so it contains approximately $2\delta$ integers, leading to the approximation $\mathbb{E}[N_I^{\text{rand}}] \approx K \cdot \frac{2\delta}{N}$.

**Part 2: Binary Attribution of Sparse Signals (BASS)**

- **Setup:** BASS maintains a search window $[\ell_k, h_k]$ of size $W_k = h_k - \ell_k$. Initially, $[\ell_0, h_0] = [0, N]$, so $W_0 = N$. The algorithm ensures that the pivotal step $c^\star$ is always within the current window: $c^\star \in [\ell_k, h_k]$. At each iteration $k$, the window size is approximately halved, so $W_k \approx N/2^k$.

- **Condition for a Guaranteed Informative Sample:** The sampled midpoint is $m_k \approx (\ell_k + h_k)/2$. The distance from this midpoint to any point in the window (including $c^\star$) is at most $W_k/2$. A sample $m_k$ is guaranteed to be in the high-information zone $I_\delta$ if this maximum possible distance is less than or equal to $\delta$. For simplicity, let's analyze the condition $|m_k - c^\star| \leqslant \delta$. This is guaranteed if $W_k/2 \leqslant \delta$, or $W_k \leqslant 2\delta$. To be conservative and match the paper's argument, we use the stricter condition that the entire search window must shrink to a size smaller than $\delta$, which is $W_k \leqslant \delta$.

- **Number of Iterations to Guarantee Hits:** We find the first iteration $k_0$ where this condition is met.

$$W_{k_0} \approx N/2^{k_0} \leqslant \delta$$
$$2^{k_0} \geqslant N/\delta$$
$$k_0 \geqslant \log_2(N/\delta).$$

Since $k_0$ must be an integer, the first iteration satisfying this is $k_0 = \lceil \log_2(N/\delta) \rceil$.

- **Constructing a Lower Bound:**

  - For the first $k_0$ iterations ($k = 1, \ldots, k_0$), we cannot guarantee the sample will be in $I_\delta$. In a worst-case analysis, we assume 0 informative samples during this phase.
  - For all subsequent iterations ($k = k_0 + 1, \ldots, K$), the condition $W_k \leqslant \delta$ holds, and every sample $m_k$ is guaranteed to be in $I_\delta$.
  - The number of such guaranteed-hit iterations is $K - k_0$. This count can't be negative, so it is $\max(0, K - k_0)$.

- **Lower Bound on Expectation:** The number of informative samples, $N_I^{\text{BASS}}$, is thus deterministically lower-bounded in this model: $N_I^{\text{BASS}} \geqslant \max(0, K - k_0)$. The expectation of a random variable is always greater than or equal to any deterministic lower bound on its value.

$$\mathbb{E}[N_I^{\text{BASS}}] \geqslant \max(0, K - k_0) = \max\left(0, K - \lceil \log_2(N/\delta) \rceil\right).$$

This completes the proof for both parts of the proposition. $\qquad\square$

