# OpenReview forum: "Binary Search for RLVR"
_ICLR.cc/2026/Conference — Submitted to ICLR 2026_

### Official Review · Reviewer_ztmB · 2025-10-28

**Soundness:** 3
**Presentation:** 3
**Contribution:** 3
**Rating:** 4
**Confidence:** 3

**Summary:**

The paper proposes a new method called BASS (Binary Attribution of Sparse Signals) to improve the utilization of sparse rewards in reinforcement learning. Specifically targeting the Reinforcement Learning from Verifiable Rewards (RLVR) paradigm—where reward signals are extremely sparse—the method reframes a single verified successful trajectory not as a single training data point, but as a blueprint. BASS performs a binary search over the trajectory’s prefixes and, at each midpoint, resamples a continuation to efficiently probe the model’s edge of competence—the boundary between success and failure. In essence, BASS is an efficient, lightweight plug-in module for RLVR that significantly enhances the exploitation of sparse reward signals, offering a novel and practical approach to sparse-reward reinforcement learning.

**Strengths:**

1. Elegant and impactful core idea: The insight—“starting from a successful trajectory and using binary search to explore the model’s reasoning boundary”—is simple yet highly insightful.

2. Solid theoretical analysis: The paper provides rigorous analysis covering both probabilistic advantages and search efficiency, demonstrating BASS’s exponential advantage over random strategies in generating high-contrast training samples.

3. Strong empirical validation: On challenging math benchmarks (AIME 24, AIME 25, and BeyondAIME), BASS consistently outperforms baselines (GRPO and Random Prefix Sampling), achieving an average improvement of +2.7 percentage points in avg@32, which convincingly validates its effectiveness.

4. High practicality: BASS is simple to implement and easily integrable into existing RLVR pipelines, making it broadly applicable.

**Weaknesses:**

1. Moderate experimental scope: Experiments are conducted only with Qwen3-8B; broader validation across diverse models and task domains is needed.

2. Idealized theoretical assumptions: The analysis assumes binary (0/1) trajectory-level rewards and a single “pivotal step” determining success—a simplification that may not hold in many real-world reasoning tasks where success is more gradual or distributed.

3. Insufficient discussion of computational trade-offs: While BASS incurs additional rollouts (K per success), the paper does not thoroughly analyze or contextualize the cost–benefit balance, especially in resource-constrained settings.

**Questions:**

See the weaknesses above

---

### Official Review · Reviewer_SQ1k · 2025-10-31

**Soundness:** 1
**Presentation:** 1
**Contribution:** 2
**Rating:** 2
**Confidence:** 3

**Summary:**

This paper proposes a method to train LLMs with rare samples efficiently in the context of reinforcement learning from verifiable rewards.

**Strengths:**

The paper addresses an important problem of sample inefficiency in reinforcement learning for LLMs, which has significant practical relevance. The proposed method of utilizing the dataset based on successful trajectories seems reasonable and promising.

**Weaknesses:**

The writing quality needs substantial improvement. The current version is not reader-friendly. For example, the paper presents the introduction, related works, and then the proposed method. I strongly suggest that the authors include a background section before introducing the method. At minimum, it should describe the problem setting, explain what RLVR is, and how GRPO works. While having a related works section is fine, the current one is not very informative. Are the related works on structured exploration, systems, and data curation recipes essential for understanding the proposed method? It is acceptable to include them, but the background I mentioned is far more crucial for helping readers understand the proposed method.

Due to the lack of background, Section 3 is difficult to follow. For instance, one may not understand what a verified trajectory refers to (in Algorithm 1).

In addition, the description of the proposed method simply explains how it works without providing motivation or illustrative examples. Furthermore, the theorems are not understandable. I tried to follow and interpret them but could not grasp their meaning. For example, in Proposition 3.3, why is the lower bound of BASS for informative samples important? It seems trivial that prioritized sampling would outperform independent sampling. Does this guarantee better LLM training? The authors also refer to GRPO here, but as mentioned earlier, without a background explanation of GRPO and related methods, it is difficult to follow.

I recommend that the authors include a figure illustrating how the proposed method works—perhaps Figure 1 could be adapted for this purpose.

Overall, the current version is far from the level expected of a top-tier AI venue manuscript. I am afraid that the necessary improvements may not be achievable within the rebuttal period. I believe the paper requires a substantial revision.

**Questions:**

See Weaknesses

---

### Official Review · Reviewer_jVdQ · 2025-10-31

**Soundness:** 1
**Presentation:** 1
**Contribution:** 1
**Rating:** 2
**Confidence:** 4

**Summary:**

The paper studies reinforcement learning from verifiable rewards (RLVR).
Given that GRPOs gradient estimates have high variance, the paper proposes a targeted data sampling strategy to reduce variance by obtaining more informative rollouts.
The paper proposes a method that performs binary search for *sequentially* generating from a prefix of a successful attempt.
Under a limited model, the paper has a statistical (not computational) comparison to randomly sampling prefixes of successful attempts.
The paper includes some results of Qwen3-8B on AIME24, AIME25, and BeyondAIME.

**Strengths:**

The paper tackles the real problem of sample-inefficient credit assignment in GRPO.
As far as I am aware the method proposed by the paper is novel.

**Weaknesses:**

* The main weakness of this paper is that it largely ignores computational efficiency. The proposed method requires *sequential* generation and evaluation of attempts. It is unclear whether / how this could be implemented efficiently.
*

* The selection of prefixes is uninformed and doesn't use other signals such as entropy, which preior work has shown to be useful. It would be valuable to compare against a more informed strategy.
* The provided anaysis in toy models is relatively straightforward and rests on unrealistic assumptions (as acknowledged in the paper).
* Line 310 highlights that GRPO fails when all attempts fail. However, it is not discussed that the method proposed by this paper does also not address / solve this problem in any way.
* The experiment in Section 4.1 is conducted in a very curated setting. Further, it does not address the concern around computational efficiency: Say we sample 8 generations in parallel with GRPO and 1 of them is successful on average (as posited in the paper). Then while sampling 8 total attempts sequentially with the method proposed in the paper, GRPO would have sampled 8 successful attempts (out of 64) in expectation.
* The reported gains on the AIME tasks are relatively small. It is unclear whether they are statistically significant.
* There is very little details on the AIME experiments. For example, what was the training time of each of the methods? How do the methods compare if training time is matched? How many GRPO training steps are completed on average for one training step of BASS?

In summary, I believe that substantial additional experiments would be needed to support the claims made in the paper.


Presentation:
* The paper is quite wordy, and seems heavily LLM-edited. The extensive use of adjectives make it quite hard to follow at times.
* Several parts of the paper are relatively redundant and could be condensed. For example, the introduction is almost a complete copy of the abstract without providing much further information / perspective.

**Questions:**

See above.

---

### Official Review · Reviewer_3kU8 · 2025-10-31

**Soundness:** 2
**Presentation:** 3
**Contribution:** 3
**Rating:** 4
**Confidence:** 4

**Summary:**

This paper addresses the RLVR setting in LLMs and proposes a method to exploit sparse successful trajectories better. Starting from a verified successful trajectory, the proposed BASS method applies a binary search over trajectory prefixes to locate the model’s “edge of competence.” This turns a single rare success into a contrastive, information-dense training group that can provide richer learning signals.

**Strengths:**

1. The paper tackles an important and timely problem in RLVR for LLMs.
2. The idea of expanding successful examples into high-signal training groups is practical and interesting.
3. The writing is clear and easy to follow.

**Weaknesses:**

1. For the theory part, the method is highly idealized and tailored to binary-search-like problems. In real LLM reasoning, failures are complex, stochastic, and often arise from multiple interdependent weaknesses. The controlled simulation (Fig. 4) effectively demonstrates that “binary search works well on a binary search problem,” but this does not justify BASS’s theoretical soundness in realistic multi-error reasoning settings. There remains a significant gap between the theory and real LLM experiments.
2. In experiments, the paper interprets BASS’s lower policy entropy (Fig. 5) as evidence of a “focused learning signal.” This interpretation conflicts with common RL theory (especially in PPO), where a rapid entropy drop usually indicates policy entropy collapse, leading to reduced exploration and premature convergence.
3. The paper didn't provide codes, which undermines reproducibility.
4. Minor: the title should avoid using acronyms.

**Questions:**

1. In real reasoning trajectories, there may be multiple unrelated failure points (e.g., a computation mistake at step 10 and a logical flaw at step 20). How would BASS behave in such cases? Would the binary search converge to only the first failure or the last success, thereby generating contrastive samples around a single boundary while ignoring other critical weak points?
2. The paper claims that BASS is “lightweight” and “simple,” but section 3.4 describes a nontrivial integration involving a “memory buffer M” and periodic batch replacement. This buffer–replacement logic adds considerable implementation overhead. Moreover, the additional (K) rollouts used by BASS are treated as a “computational investment.” If competing baselines such as GRPO or RPS were allowed to use the same total computation budget (e.g., increasing group size from 8 to (8+K) or training for more steps), would BASS still maintain its reported +2.7 pp improvement?

---

### Meta-Review · Area_Chair_k67i · 2025-12-23

**Summary:**

The reviewers noted the importance of the problem tackled in this paper. However, they agreed that the writing and the paper organization make the contributions hard to understand. In addition, they raised the issues of a too-idealized setting for the theoretical analysis and of a too-limited experimental evaluation.

**Reviewer Concerns:**

There was no rebuttal.

**Reviewer Scores:**

There was no rebuttal.

---

### Decision · Program_Chairs · 2026-01-26

Reject